# Wide-Range Flexible Capacitive Pressure Sensors Based on Dielectrics with Various Porosity

**DOI:** 10.3390/mi13101588

**Published:** 2022-09-25

**Authors:** Huiyang Yu, Chengxi Guo, Xin Ye, Yifei Pan, Jiacheng Tu, Zhe Wu, Zefang Chen, Xueyang Liu, Jianqiu Huang, Qingying Ren, Yifeng Li

**Affiliations:** 1College of Computer Science and Technology, Nanjing Tech University, Nanjing 211816, China; 2Key Laboratory of MEMS of the Ministry of Education, Southeast University, Nanjing 210096, China; 3College of Electronic and Optical Engineering & College of Flexible Electronic (Future Technology), Nanjing University of Posts and Telecommunication; Nanjing 210023, China

**Keywords:** wide-range, capacitive, pressure sensor, various porosity

## Abstract

Wide-range flexible pressure sensors are in difficulty in research while in demand in application. In this paper, a wide-range capacitive flexible pressure sensor is developed with the foaming agent ammonium bicarbonate (NH_4_HCO_3_). By controlling the concentration of NH_4_HCO_3_ doped in the polydimethylsiloxane (PDMS) and repeating the curing process, pressure-sensitive dielectrics with various porosity are fabricated to expand the detection range of the capacitive pressure sensor. The shape and the size of each dielectric is defined by the 3D printed mold. To improve the dielectric property of the dielectric, a 1% weight ratio of multi-walled carbon nanotubes (MWCNTs) are doped into PDMS liquid. Besides that, a 5% weight ratio of MWCNTs is dispersed into deionized water and then coated on the electrodes to improve the contact state between copper electrodes and the dielectric. The laminated dielectric layer and two electrodes are assembled and tested. In order to verify the effectiveness of this design, some reference devices are prepared, such as sensors based on the dielectric with uniform porosity and a sensor with common copper electrodes. According to the testing results of these sensors, it can be seen that the sensor based on the dielectric with various porosity has higher sensitivity and a wider pressure detection range, which can detect the pressure range from 0 kPa to 1200 kPa and is extended to 300 kPa compared with the dielectric with uniform porosity. Finally, the sensor is applied to the fingerprint, finger joint, and knee bending test. The results show that the sensor has the potential to be applied to human motion detection.

## 1. Introduction

With the deepening research in the field of Artificial Intelligence (AI), it creates higher requirements for flexible sensors [1,2,3,4,5,6,7]. Among all these sensors, pressure sensors have been widely concerned because of their wide application [8]. In addition, due to their wide application, it is necessary to continuously improve the sensitivity of pressure sensors to adapt to more and more high-end applications [9,10]. On some occasions, it requires that the sensors have a wide pressure detection range. Taking the electronic skin as an example, it not only needs to be able to sense the soft touch of feathers but also needs to be able to sense the hit of fists. There have been some wide-range pressure sensors reported over the years [11,12,13,14,15,16,17,18,19,20,21,22,23], however, those are mostly resistive types, in which the resistance of the sensor varies with applied pressure [13,14]. Although resistive pressure sensors with a wide detection range were most reported, they have an obvious disadvantage of high power consumption [15,16]. This feature is very negative for many portable devices [17] because a long standby time is usually necessary [18,19]. Conversely, a capacitive pressure sensor has much lower power consumption and fast response [24,25]. Therefore, it is necessary to explore flexible capacitive pressure sensors with high sensitivity and a wide detection range.

Recently, many reported research findings have successfully realized high sensitivity in many flexible capacitive pressure sensors [26,27,28,29,30]. For example, some attempts have been made to fabricate micro nanostructures to improve the sensitivity of sensors, but these sensors mostly have high sensitivity only in a relatively low pressure [31,32,33,34]. At present, some reported capacitive pressure sensors are fabricated based on a bionic mechanism [35,36]. Besides that, the sacrificial mold method is also used to fabricate porous microstructure [35,36,37,38]. Using salt particles or sugar particles as the sacrificial material to fabricate a porous structure can achieve high sensitivity and a stable production process, but it may cause some residues in the pressure-sensitive layer. Therefore, it is necessary to develop a simple, efficient, environment-friendly and performance-friendly method of fabricating a pressure-sensitive dielectric.

In this paper, a flexible capacitive pressure sensor is proposed. With regard to the presently reported flexible sensors, the pressure-sensitive layer is fabricated to be a porous material. However, it is typical that the pressure-sensitive dielectric layer has various porosity. In this sensor, PDMS is used as the flexible substrate, which is a very promising flexible polymer material because it can meet the requirements of linear response in a wide range [39,40,41]. NH_4_HCO_3_ is chosen here because it is not only economically efficient and environmentally-friendly, but also can be completely decomposed under heating conditions [42]. Besides that, the doping concentration and the porosity of dielectrics can be adjusted according to the actual need. In addition, by controlling the amount of doped NH_4_HCO_3_ in different layers, porous materials with various porosity can be fabricated to broaden the detecting range of the sensors. Therefore, it is a potential method for designing a wide range of flexible pressure sensors.

## 2. Materials and Methods

### 2.1. The Prototype of the Sensor

The structural diagram of the sensor is shown in Figure 1a. The sensor consists of three parts, two copper electrodes and a pressure-sensitive dielectric layer. The most remarkable feature of this sensor is that the pressure-sensitive dielectric layer has varied porosity; this design is to enlarge the pressure detection range of the sensor. The working principle is shown in Figure 1b. The dielectric could be considered as two spring damper elements connected in series. The dielectric part with higher porosity corresponds to the spring-damper element with a small elastic coefficient, the dielectric part with lower porosity corresponds to the spring-damper element with a large elastic coefficient. When the pressure is in a low range, the dielectric part with higher porosity plays a primary role in response and the capacitance increase with the applied pressure. As the pressure becomes larger, the dielectric part with lower porosity and the capacitance of the sensor can be further increased with larger pressure. With this design, the detection range of the sensor can be obviously enlarged.

### 2.2. The Fabrication Process of the Sensor

The fabrication process of the sensors is shown in Figure 2: (a) the mold (prepared by the 3D printer) and materials (2 wt%, 3 wt%, 4 wt%, 6 wt% NH_4_HCO_3_, and 1 wt% MWCNTs) are prepared. NH_4_HCO_3_ particles are put into the Kibbler machine and fully ground to a uniform fine powder. (b) Four groups of additives (3 wt% NH_4_HCO_3_ and 1 wt% MWCNTs, 4 wt% NH_4_HCO_3_ and 1 wt% MWCNTs, 2 wt% NH_4_HCO_3_ and 1 wt% MWCNTs, 6 wt% NH_4_HCO_3_, and 1 wt% MWCNTs) are added to four groups of PDMS liquid and fully stirred with a glass rod for 5 min. (c) A release agent is sprayed on the surface inside the mold and then the completely mixed solution is dropped into the grid in the mold; the size of each grid is 15 mm × 15 mm × 6 mm. (d) The mold is put into the vacuum pump to remove the air introduced during solution mixing. For the multilayer dielectric, only half volume of PDMS solution is added in this step. (e) The mold is heated in the oven at 80 °C for 2 h. (f) For multilayer dielectric fabricating: PDMS solution with other portion has to be added again on the previously cured layer after removing bubbles. (g) Then it is put into the oven and heated again for 2 h. After curing, the top and bottom part of the pressure-sensitive dielectrics are closely pasted together. The pressure-sensitive material with various porosity is obtained. (h) After the cured dielectrics cool down, the dielectric is pulled out of the mold with tweezers. (i) Preparation of carbon electrodes: MWCNTs with a mass fraction of 5% are dispersed into deionized water to form suspensions. Ultrasonic oscillation machine and 100 °C hot water are used for 20 min to obtain a uniformly dispersed carbon nanotube suspension. Fabricated porous dielectric is immersed in carbon nanotube suspension until the surface is completely adsorbed by carbon black. Then the sample is put into the oven and dried at 70 °C. After the deionized water has completely evaporated, the pressure-sensitive material was taken out. The MWCNTs stuck on the surface of the inner side of the electrodes and electrodes with microstructure are obtained. (j) The two MWCNTs treated electrodes are attached to the dielectric layer.

### 2.3. The Pictures of the Fabricated Dielectrics and the Electrodes

The optical microscope was adopted to observe the fabricated dielectrics and the copper electrodes. The pictures are shown in Figure 3. The surface of pure solid PDMS and porous PDMS is shown in Figure 3a,b. It can be seen that pure PDMS surface is much flatter than the porous PDMS and some holes can be viewed from the surface of the porous PDMS. The pictures of the copper electrode and the copper electrode treated with MWCNTs are shown in Figure 3c,d. Comparing these two pictures, it can be seen that the electrode treated with MWCNTs is much rougher than the electrode not treated because many small bumps are formed on the surface of the former. The fabricated dielectric is shown in Figure 3e. The cross-section view of it is shown in Figure 3f, from which it can be seen that there is an obvious boundary on the cross-section of the dielectric. The sample exhibits different porosity on both sides of the boundary.

## 3. Results and Discussion

### 3.1. The Finite Element Simulation Results of the Dielectrics

To verify the dielectric with various porosity is effective in improving the performance of the sensor, the finite element simulation is carried out to analyze the mechanical properties of these three kinds of dielectrics, which are pure PDMS, PDMS with 10% uniform porosity and PDMS with various porosity, respectively. The half volume of the third kind of dielectric is with 5% porosity, and the other part is with 15% porosity, so the average porosity is the third dielectric and is the same as the second dielectric. The deformation of the dielectric at 250 kPa and 1000 kPa pressure is analyzed and the simulation results are shown in Figure 4. According to these results, the dielectric with various porosity shows the largest deformation, which means that the dielectric with various porosity is more sensitive to pressure and can realize a larger dynamic range.

### 3.2. The Testing System and Results of the Dielectrics

The fabricated sensors are tested with the instruments shown in Figure 5a. The pressure testing instrument is used to apply the pressure on the sensor and measure the applied pressure. The portable LCR Meter is used to test the capacitance of the sensor. Four sensors are tested, the dielectric of which are pure PDMS, PDMS with uniform porosity doping 3 wt% NH_4_HCO_3_, PDMS with uniform porosity doped 4 wt% NH_4_HCO_3_, PDMS with various porosity doped 2 wt% and 6 wt% NH_4_HCO_3_, respectively. It has to be pointed out that 1 wt% MWCNTs is another additive added in the dielectrics doped NH_4_HCO_3_, which is selected to enhance the strength and dielectric property of dielectrics for its large aspect ratio and excellent electrical characteristic. When the doped MWCNTs are wrapped by the PDMS in the flexible dielectric, they could be considered to be electric dipoles under an external electric field, which could enhance the equivalent polarized electric field of the flexible dielectric. When pressure is applied on the sensor, the electric dipoles get closer and their coupling effect becomes stronger. As a result, the dielectric properties of the flexible dielectric can be improved to some extent. When the pressure is applied, the capacitance of the sensors increases and the results are shown in Figure 5b. For the sensors with a dielectric made of pure PDMS and PDMS with uniform porosity, the sensors’ sensitivity increases with the sensor’s porosity and the limitation of the applied pressure is 900 kPa. When the pressure is larger than 900 kPa, the sensor is too hard to respond to the increased pressure. While for the sensor with various porosity, it can respond to the pressure range from 0 kPa–1200 kPa and has the highest sensitivity compared with the other three devices. In general, the sensor dielectric which has varied porosity shows the highest sensitivity and widest pressure detection range. The intrinsic reason is that the component of the dielectric with larger porosity is more sensitive to low pressure. The component of the dielectric with lower porosity has a larger Young’s Modulus and can sustain larger pressure compared to the dielectric with uniform porosity.

To verify the effect of the MWCNTs on dielectrics, two sensors were fabricated, the dielectric of which is PDMS doped 4 wt% NH_4_HCO_3_ and PDMS doped 4 wt% NH_4_HCO_3_ and 1 wt% MWCNTs, respectively. Transient pressure was applied three times on these two devices. The capacitance variation with pressure is recorded and the results are shown in Figure 5c. From these results, it can be seen that the sensor with dielectrics doped both NH_4_HCO_3_ and MWCNTs has higher sensitivity, which was improved about 20% compared with the sensor with a dielectric without doping MWCNTs.

Besides that, the sensors with common copper foil electrodes and the senor with electrodes treated with MWCNTs are tested, the dielectric of these two sensors belongs to the kind with various porosity. The testing results are shown in Figure 5d. According to these testing results, the sensor with electrodes treated with MWCNTs is with larger sensitivity. On the basis of the optical microscope pictures shown in Figure 3c,d, electrodes treated with MWCNTs is rougher than the common copper foil. It will have two effects. On the one hand, the rough electrode surface will bring about better electrical contact with the dielectric and then better transfer the polarization state in the dielectric. On the other hand, the electrode with a rough surface has a larger equivalent area than the electrode with flat surfaces, and the small bumps on the surface of the electrodes treated with MWCNTs are also sensitive to applied pressure. Based on these two factors, the sensitivity of the sensor with electrodes treated with MWCNTs is a little higher than the sensor that is not treated with MWCNTs.

The hysteresis property of the sensor is investigated. The tested sensor is the one based on the dielectric with various porosity. The tested results are shown in Figure 5e. In the pressing cycle, the applied pressure is increased from 0 kPa to 1200 kPa. In the releasing cycle, the applied pressure is decreased from 1200 kPa to 0 kPa. According to the testing results, the hysteresis error of the tested sensor is about 5%. Besides that, the repeatability property of the sensor is also investigated. The applied pressure is periodically increased from 0 kPa to 1200 kPa. The capacitance variation of the sensors is recorded in the first cycle, the 50th cycle, and the 200th cycle. The capacitance variation of the sensor is shown in Figure 5f. From these results, it can be seen that the repeatability error of the sensor is about 6% and it mainly occurred in the first 50 cycles. The sensitivity degrade is obviously smaller in the following 150 cycles. It could be concluded that the sensitivity degrade will not last forever and the reason for this degrade in the first 50 cycles is probably that the dielectric, electrodes, and the bonding state between them have not yet reached stability. Therefore, to improve the reliability of devices, the aging experiments should be carried out for newly fabricated devices.

### 3.3. The Application Testing Results of the Sensors

As an application, the capacitive sensor is applied for finger pressing, finger joint bending, and knee joint bending test. In the finger pressing test, the volunteer’s finger pressed on the top of the sensor with a speed of twice a second, the capacitance variation of the sensor was recorded with an LCR meter, and the results are shown in Figure 6a. According to the testing result, the sensor can respond quickly to the applied pressure. The peak value of the curve corresponds to the state in which pressure is applied to the sensor. The valley value of the curve corresponds to the state in which pressure is released. The peak value of the curve fluctuates somewhat because the pressure applied by the fingers cannot be kept exactly the same every time. However, the valley value of the curve is almost constant. It is because the sensor can return to a stable capacitance value when the pressure is released as long as it has good repeatability. As a result, it proves that the tested sensor has good repeatability from a certain point of view.

In the finger joint bending test, the sensor was stuck to the joint at the base of the finger. Then the joint switches between bending and stretching. The capacitance variation is recorded and the results are shown in Figure 6b. The peak value of the curve corresponds to the state in which the joint is bending. The valley value of the curve corresponds to the state in which the joint is stretching. From the testing results, it can be seen that the sensor can respond timely to the movement of the finger joint.

In the knee joint bending test, the sensor is stuck to the knee joint and the volunteer repeat squatting and standing movements. The testing results are shown in Figure 6c. The peak value of the curve corresponds to the state in which the knee bends down. The valley value of the curve corresponds to the state in which the knee stretches. From the testing results, it can be seen that the sensor can respond timely to the movement of the knee movement.

In summary, according to the testing results shown in Figure 6, the flexible capacitive pressure sensors based on the dielectrics with various porosity works well in the detection of human movement and thus have a great potential to be applied in the fields of human motion monitoring and wearable devices.

A comparison of some reported capacitive pressure sensors and our work has been listed in Table 1. Our device has a wide detection range and fast response time compared with most of these sensors.

## 4. Conclusions

In this paper, a flexible capacitive pressure sensor is fabricated, which consists of a multilayer pressure-sensitive dielectric, the top and bottom copper foil electrodes. The mold used to fabricate dielectrics is prepared by a 3D printing machine. The foaming agent NH_4_HCO_3_ is used to fabricate porous dielectrics. In order to improve the performance of the capacitive sensor, a small amount of MWCNTs is also doped in the dielectric layer and mounted on the electrodes of the sensor. According to the testing results, the dielectric with various porosity can detect the pressure in the range of 0–1200 kPa and extend 300 kPa compared with the sensor based on the dielectric layer with uniform porosity. Finally, the corresponding human joint motion test is carried out on the pressure sensor with various porosity in the dielectric, which proves that it has the potential to be applied in the field of wearable devices.

## Figures and Tables

**Figure 1 micromachines-13-01588-f001:**
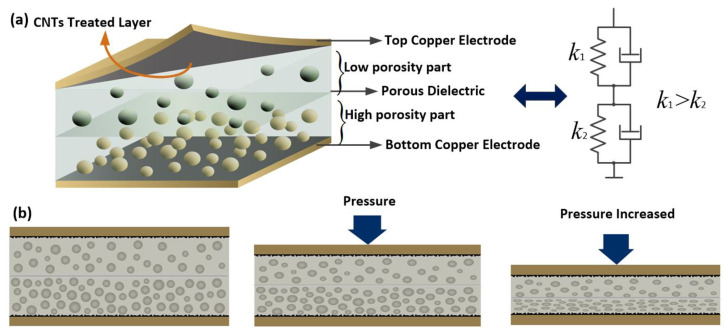
The structural diagram and the working principle of the sensor: (**a**) the structural diagram of the sensor; (**b**) the cross-section view and the working principle of the sensor.

**Figure 2 micromachines-13-01588-f002:**
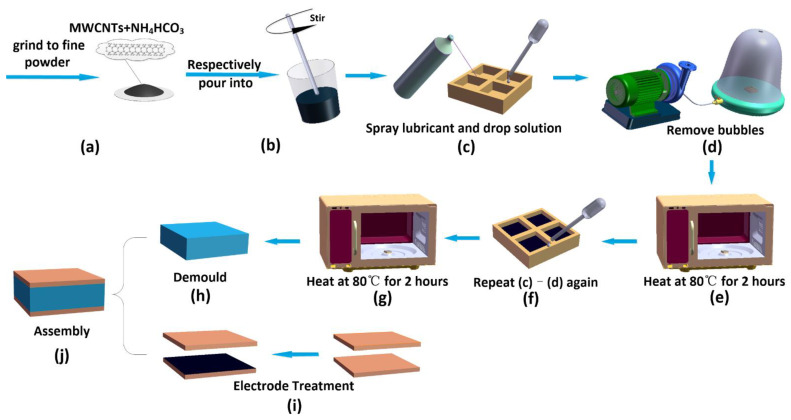
Design and fabrication of the flexible capacitive sensor: (**a**) Prepare the MWCNTs and NH_4_HCO_3_ powder. (**b**) Mix the powder with PDMS. (**c**) Spray the release agent and pour into the mixture into the mold. (**d**) Vacuum to remove air. (**e**) Heat and solidify the mixture. (**f**) Repeat step (**c**) and step (**d**). (**g**) Heat and solidify the mixture again. (**h**) Release the dielectric from the mold. (**i**) Electrode treatment. (**j**) Assembly the dielectric and the electrodes.

**Figure 3 micromachines-13-01588-f003:**
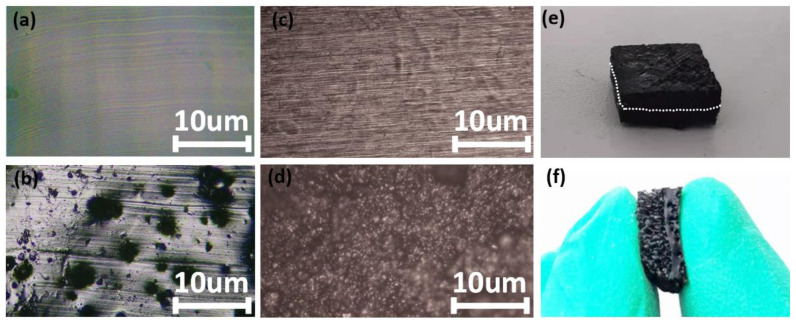
The optical microscope pictures of the fabricated dielectrics and copper electrodes: (**a**) the surface of pure PDMS. (**b**) The surface of porous PDMS doped 4 wt% NH_4_HCO_3_. (**c**) The surface of the copper electrode. (**d**) The surface of the copper electrode treated with MWCNTs. (**e**) The picture of fabricated dielectrics. (**f**) The cross-section view of the fabricated dielectric.

**Figure 4 micromachines-13-01588-f004:**
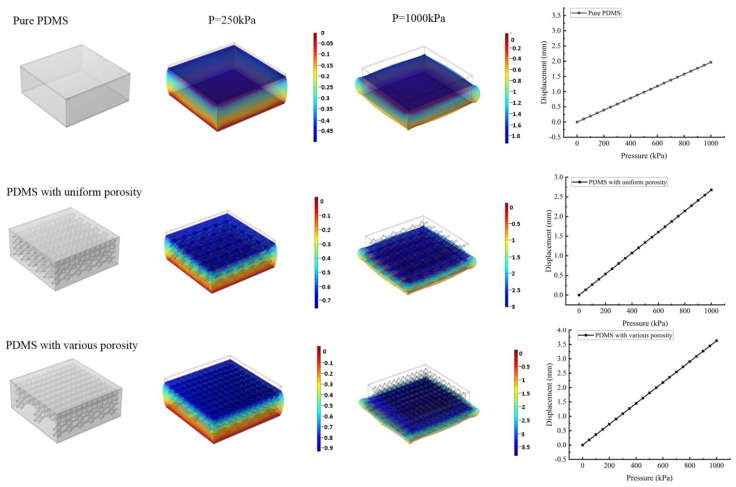
The finite simulation results of dielectrics with different porosity.

**Figure 5 micromachines-13-01588-f005:**
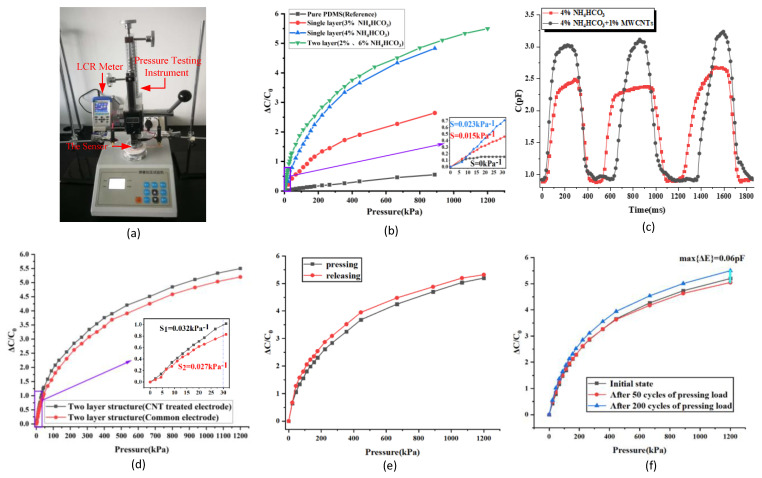
The testing system and the testing results: (**a**) the testing system. (**b**) The sensitivity of the sensors. (**c**) The sensitivity of the sensors with copper electrodes and electrodes treated with CNTs. (**d**) The press testing of the sensors with dielectric doped CNTs and not doped CNTs. (**e**) The hysteresis test results of the sensor. (**f**) The repeatability test results of the sensor.

**Figure 6 micromachines-13-01588-f006:**
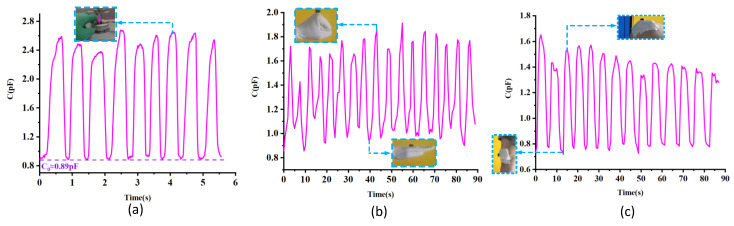
The application of the sensor: (**a**) the testing results of the finger pressing. (**b**) The testing results of the finger joint. (**c**) The testing results of the knee joint.

**Table 1 micromachines-13-01588-t001:** Comparison of performance of different capacitive sensors.

Electrodes	Dielectric	Sensitivity (Dynamic Range)	Response Time (ms)	Ref.
PDMS/Ag Nanowire	PDMS/ZnO	0.0178 (0–16 kPa) 0.0057 (16–100 kPa)	130 (loading) 190 (unloading)	[43]
Graphite	PDMS	0.62 (0–2 kPa) 0.28 (2–6 kPa) 0.06 (6–10 kPa)	200 (loading) 400 (unloading)	[44]
Indium oxide/Polyethylene terephthalate (PET)	PDMS	0.01 (0–200 kPa) 0.0009 (200–1000 kPa)	15 (loading)	[45]
Ag Nanowire/PEDOT:PSS	Ion-gel film	0.32 (1 Pa–10 kPa) 0.07 (10–50 kPa)	227 (loading) 232 (unloading)	[46]
Cu	PDMS and carbon conductive paste	1.1 (4 Pa–10 kPa) 0.4 (10–100 kPa)	60 (loading) 120 (unloading)	[47]
Copper and MWCNTs	Porous PDMS and MWCNTs	0.0125 (0–200 kPa) 0.003 (200–1200 kPa)	100 (loading) 100 (unloading)	Our work

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
