# Peer review of "Wide-Range Flexible Capacitive Pressure Sensors Based on Dielectrics with Various Porosity"

_micromachines, 2022, doi:10.3390/mi13101588_

Round 1

Reviewer 1 Report

In this manuscript, the authors developed a wide-range capacitive flexible pressure sensor with the foaming agent ammonium bicarbonate (NH4HCO3). By controlling the concentration of NH4HCO3 doped in 15 the polydimethylsiloxane (PDMS) and repeat the curing process, pressure-sensitive dielectrics with 16 various porosity are fabricated to expand the detection range of capacitive pressure sensor. A wide pressure detection range, which can detect the pressure range from 26 0 kPa to 1200 kPa is achieved. The analysis made in the manuscript is clear for understanding the subject. This work is recommended to be published before the following observations are addressed.

1)      In figure 1, working principles are recommended to be illustrated more clearly.

2)      Scale bars should be added in Figure 3.

3)      Scale bars in Figure 4 is not clear.

4)      Repeatability test of the sensor is highly recommended.    

5)      Various flexible pressure sensors based on different mechanisms have been reported. The authors should compare the current results with the reported work. In addition, the authors may refer to related publications: 1. Adv. Intell. Syst., 4: 2200050. https://doi.org/10.1002/aisy.202200050; 2. IEEE Robotics and Automation Letters, 2022, 7(2): 5127-5134; 3. Advanced Materials Technologies, 2021, 6(11): 2100616.

Author Response

Reply in general: Firstly, we want to express our high respect to the reviewer for his/her patience, carefulness and intelligence in helping us to improve this paper. Secondly, the comments will be replied as follows:

1)      In figure 1, working principles are recommended to be illustrated more clearly.

Reply: An equivalent mechanical model of the pressure-sensitive dielectric has been supplemented. The working principles of the device has been explained in more detail. The dielectric with different porosity could be considered as springs with different elastic constants. The component with higher porosity corresponds to the spring with larger elastic constant. The component with lower porosity corresponds to the spring with smaller elastic constant and they are connected in series. When the pressure is in a low range, the dielectric part with higher porosity plays a primary role in response and the capacitance increase with the applied pressure. As the pressure becomes larger, the dielectric part with lower porosity and the capacitance of the sensor can be further increased with larger pressure.

2)      Scale bars should be added in Figure 3.

Reply: Figure 3 has been replaced by the pictures with scale bars.

3)      Scale bars in Figure 4 is not clear.

Reply: Figure 4 has been replaced.

4)      Repeatability test of the sensor is highly recommended.

Reply: The repeatability test of the sensor has been included in the manuscript shown in Figure 5(f).

5)      Various flexible pressure sensors based on different mechanisms have been reported. The authors should compare the current results with the reported work. In addition, the authors may refer to related publications: 1. Adv. Intell. Syst., 4: 2200050. https://doi.org/10.1002/aisy.202200050; 2. IEEE Robotics and Automation Letters, 2022, 7(2): 5127-5134; 3. Advanced Materials Technologies, 2021, 6(11): 2100616.

Reply: We have added these related literatures in the revised manuscript.

Reviewer 2 Report

In this manuscript, the authors developed a wide range flexible capacity sensor based on porous PDMS and multi walled carbon nanotubes (MWCNTs). The sensor material preparation, architecture design, simulation modeling, testing performance and potential application demo were included in the manuscript. Overall, it is an interesting study and could be able to inspire the future pressure sensor development in the industry. However, I have the following comments and suggestions and hope the authors could address:

1. Line [33], The claim "With the deepening research in the field of Artificial Intelligence (AI), it puts forward higher requirements for flexible sensors [1-4]" needs more elaboration. References [1-4] only talk about sensors. No strong connection between AI and flexible sensor with the current expression.

2. Line [38], please help indicate the scenarios that need wide range flexible sensors and add references.

3. Figure 4. The heat maps and color bars are difficult to tell the difference between the 3 configurations. Maybe curve plotting can help the readers understand the performance differences better.

4. The authors utilized MWCNTs to improve the performance of sensor in two aspects. Mixing it with the PDMS and surface treating the electrodes. Could the authors elaborate more on what is the mechanism of adding MWCNTs for performance improvement? For the electrode surface treatment, I am not quite convinced, since we may not need perfect electrical contact for capacitance-based sensor and the small bump seems too tricky for me to believe it can be stable and repeatable.

5. From the repeating pressure load cycles testing, the performance degrades 6% after 200 cycles. What is the state-of-arts degradation for resistance-based sensors? How does the authors improve the reliability for real industry applications?

6. Could the authors provide the statistical performance analysis of the same batch of many sensors? If the performance variance is large, calibration will be needed to each individual sensors before shipping. This is a tedious process for real sensor.

7. It would be great if the authors could summarize the performance of previous related works and compare with the study in this manuscript in a table on sensitivity, detection range, reliability etc. 

Author Response

Reply in general: Firstly, we want to express our high respect to the reviewer for his/her patience, carefulness and intelligence in helping us to improve this paper. Secondly, the comments will be replied as follows:

  1. Line [33], The claim "With the deepening research in the field of Artificial Intelligence (AI), it puts forward higher requirements for flexible sensors [1-4]" needs more elaboration. References [1-4] only talk about sensors. No strong connection between AI and flexible sensor with the current expression.

Reply: We have added some references with stronger relevance.

  1. Line [38], please help indicate the scenarios that need wide range flexible sensors and add references.

Reply: We have added the scenarios in need of wide range flexible sensors. In fact, pressure sensor is the kind of sensor with very wide application and the requirements for the detection range of the sensor are various in different application situations. Take the electronic skin for example, it not only needs to be able to sense the soft touch of feathers but also needs to be able to sense the hit of fists.

  1. Figure 4. The heat maps and color bars are difficult to tell the difference between the 3 configurations. Maybe curve plotting can help the readers understand the performance differences better.

Reply: Curve plotting has been added in Figure 4.

  1. The authors utilized MWCNTs to improve the performance of sensor in two aspects. Mixing it with the PDMS and surface treating the electrodes. Could the authors elaborate more on what is the mechanism of adding MWCNTs for performance improvement? For the electrode surface treatment, I am not quite convinced, since we may not need perfect electrical contact for capacitance-based sensor and the small bump seems too tricky for me to believe it can be stable and repeatable.

Reply: Doping is a commonly used method to improve the electric or dielectric property of the flexible substrate. Take PDMS for example, when the doped MWCNTs are wrapped by the PDMS, they could be considered as electric dipoles, which can enhance the equivalent polarized electric field in the flexible dielectric. And when the pressure is applied on the sensor, the electric dipoles get closer and their coupling effect becomes stronger. As a result, the dielectric properties of the flexible dielectric can be improved to some extent. For electrodes, the MWCNTs is used to form the rough surface on the electrode surface. In one hand, the rough electrode surface can form good contact with the dielectric. In the other hand, the electrode with rough surface has a greater equivalent surface area than the electrode with flat surface. Besides that, the small bumps on the surface of the electrodes are also sensitive to applied pressure. In some reported research work, silver nanowires and graphene are often used as electrodes of capacitive pressure sensors, some related references are as follows:

[1]S, Chen.; X, Guo. Improving the Sensitivity of Elastic Capacitive Pressure Sensors Using Silver Nanowire Mesh Electrodes. IEEE Transactions on Nanotechnology. 2015, 14(4), 619-623.

[2] C, Han.; B, Park.; M.S, Oh.; S, Jung.; J, Kim. Photo-induced fabrication of Ag nanowire circuitry for invisible, ultrathin, conformable pressure sensors. J. Mater. Chem. C. 2017, 5, 9986-9994.

[3] An, B.W.; S, Heo.; S, Ji.; F, Bien.; J, Park. Transparent and flexible fingerprint sensor array with multiplexed detection of tactile pressure and skin temperature. Nat Commun. 2018, 9, 2458.

[4] Yang, J.; Luo, S.; Zhou, X.; Li, J.; Fu J.; Yang, W.; Wei, D. Flexible, Tunable, and Ultrasensitive Capacitive Pressure Sensor with Microconformal Graphene Electrodes. ACS Appl Mater Interfaces. 2019, 11(16), 14997-15006.

  1. From the repeating pressure load cycles testing, the performance degrades 6% after 200 cycles. What is the state-of-arts degradation for resistance-based sensors? How does the authors improve the reliability for real industry applications?

Reply: Although the performance degrades after 200 cycles, this degrade mainly occurred in the first 50 cycles and this degrade is not very obvious in the following 150 cycles according to the load cycles testing shown in Figure 5(f). It could be concluded that this degrade will not last too long. According to our estimation, the reason for this sensitivity degrade in the first 50 cycle is probably that the dielectric, electrodes and the bonding state between them have not yet reached stability. Therefore, to improve the reliability of devices, the aging experiments could be carried out for newly fabricated devices.

  1. Could the authors provide the statistical performance analysis of the same batch of many sensors? If the performance variance is large, calibration will be needed to each individual sensors before shipping. This is a tedious process for real sensor.

Reply: About the consistency of the sensor, we once fabricated three capacitors with the size of   5mm×5mm×2mm. Their statistical performance is analyzed and the testing results are shown in the following figure. Because the size of these three samples is not the same with the device exhibited in the manuscript, these results is not displayed in the manuscript. Because the thickness of these three samples is one third of the sensor exhibited in the manuscript, the sensor sensitivity is decreased to one-third of the sensor exhibited in the manuscript. However, the statistical performance of these three samples is similar according to the testing results. Therefore, the consistency of this sensor is not bad and this method is expected to be applied in batch fabrication.

  1. It would be great if the authors could summarize the performance of previous related works and compare with the study in this manuscript in a table on sensitivity, detection range, reliability etc. 

Reply: The summarize of the previous related works and the study in this manuscript on sensitivity, detection range has been added in the manuscript.

Round 2

Reviewer 1 Report

The manuscript could be accepted as it is.